# Optimization, Learning, and Games with Predictable Sequences

**Alexander Rakhlin**
University of Pennsylvania

**Karthik Sridharan**
University of Pennsylvania

## Abstract

We provide several applications of Optimistic Mirror Descent, an online learning algorithm based on the idea of predictable sequences. First, we recover the Mirror Prox algorithm for offline optimization, prove an extension to Hölder-smooth functions, and apply the results to saddle-point type problems. Next, we prove that a version of Optimistic Mirror Descent (which has a close relation to the Exponential Weights algorithm) can be used by two strongly-uncoupled players in a finite zero-sum matrix game to converge to the minimax equilibrium at the rate of $\mathcal{O}((\log T)/T)$. This addresses a question of Daskalakis et al [6]. Further, we consider a partial information version of the problem. We then apply the results to convex programming and exhibit a simple algorithm for the approximate Max Flow problem.

## 1   Introduction

Recently, no-regret algorithms have received increasing attention in a variety of communities, including theoretical computer science, optimization, and game theory [3, 1]. The wide applicability of these algorithms is arguably due to the black-box regret guarantees that hold for arbitrary sequences. However, such regret guarantees can be loose if the sequence being encountered is not "worst-case". The reduction in "arbitrariness" of the sequence can arise from the particular structure of the problem at hand, and should be exploited. For instance, in some applications of online methods, the sequence comes from an additional computation done by the learner, thus being far from arbitrary.

One way to formally capture the partially benign nature of data is through a notion of predictable sequences [11]. We exhibit applications of this idea in several domains. First, we show that the Mirror Prox method [9], designed for optimizing non-smooth structured saddle-point problems, can be viewed as an instance of the predictable sequence approach. Predictability in this case is due precisely to smoothness of the inner optimization part and the saddle-point structure of the problem. We extend the results to Hölder-smooth functions, interpolating between the case of well-predictable gradients and "unpredictable" gradients.

Second, we address the question raised in [6] about existence of "simple" algorithms that converge at the rate of $\tilde{\mathcal{O}}(T^{-1})$ when employed in an uncoupled manner by players in a zero-sum finite matrix game, yet maintain the usual $\mathcal{O}(T^{-1/2})$ rate against arbitrary sequences. We give a positive answer and exhibit a fully adaptive algorithm that does not require the prior knowledge of whether the other player is collaborating. Here, the additional predictability comes from the fact that both players attempt to converge to the minimax value. We also tackle a partial information version of the problem where the player has only access to the real-valued payoff of the mixed actions played by the two players on each round rather than the entire vector.

Our third application is to convex programming: optimization of a linear function subject to convex constraints. This problem often arises in theoretical computer science, and we show that the idea of

predictable sequences can be used here too. We provide a simple algorithm for $\epsilon$-approximate Max Flow for a graph with $d$ edges with time complexity $\tilde{\mathcal{O}}(d^{3/2}/\epsilon)$, a performance previously obtained through a relatively involved procedure [8].

## 2 Online Learning with Predictable Gradient Sequences

Let us describe the online convex optimization (OCO) problem and the basic algorithm studied in [4, 11]. Let $\mathcal{F}$ be a convex set of moves of the learner. On round $t = 1, \ldots, T$, the learner makes a prediction $f_t \in \mathcal{F}$ and observes a convex function $G_t$ on $\mathcal{F}$. The objective is to keep *regret* $\frac{1}{T} \sum_{t=1}^{T} G_t(f_t) - G_t(f^*)$ small for any $f^* \in \mathcal{F}$. Let $\mathcal{R}$ be a 1-strongly convex function w.r.t. some norm $\| \cdot \|$ on $\mathcal{F}$, and let $g_0 = \arg\min_{g \in \mathcal{F}} \mathcal{R}(g)$. Suppose that at the beginning of every round $t$, the learner has access to $M_t$, a vector computable based on the past observations or side information. In this paper we study the **Optimistic Mirror Descent** algorithm, defined by the interleaved sequence

$$f_t = \underset{f \in \mathcal{F}}{\arg\min} \ \eta_t \langle f, M_t \rangle + \mathcal{D}_{\mathcal{R}}(f, g_{t-1}) \ , \ g_t = \underset{g \in \mathcal{F}}{\arg\min} \ \eta_t \langle g, \nabla G_t(f_t) \rangle + \mathcal{D}_{\mathcal{R}}(g, g_{t-1}) \quad (1)$$

where $\mathcal{D}_{\mathcal{R}}$ is the Bregman Divergence with respect to $\mathcal{R}$ and $\{\eta_t\}$ is a sequence of step sizes that can be chosen adaptively based on the sequence observed so far. The method adheres to the OCO protocol since $M_t$ is available at the beginning of round $t$, and $\nabla G_t(f_t)$ becomes available after the prediction $f_t$ is made. The sequence $\{f_t\}$ will be called primary, while $\{g_t\}$ – secondary. This method was proposed in [4] for $M_t = \nabla G_{t-1}(f_{t-1})$, and the following lemma is a straightforward extension of the result in [11] for general $M_t$:

**Lemma 1.** *Let $\mathcal{F}$ be a convex set in a Banach space $\mathcal{B}$. Let $\mathcal{R} : \mathcal{B} \to \mathbb{R}$ be a 1-strongly convex function on $\mathcal{F}$ with respect to some norm $\| \cdot \|$, and let $\| \cdot \|_*$ denote the dual norm. For any fixed step-size $\eta$, the Optimistic Mirror Descent Algorithm yields, for any $f^* \in \mathcal{F}$,*

$$\sum_{t=1}^{T} G_t(f_t) - G_t(f^*) \le \sum_{t=1}^{T} \langle f_t - f^*, \nabla_t \rangle$$

$$\le \eta^{-1} R^2 + \sum_{t=1}^{T} \| \nabla_t - M_t \|_* \| g_t - f_t \| - \frac{1}{2\eta} \sum_{t=1}^{T} \left( \| g_t - f_t \|^2 + \| g_{t-1} - f_t \|^2 \right) \quad (2)$$

*where $R \ge 0$ is such that $\mathcal{D}_{\mathcal{R}}(f^*, g_0) \le R^2$ and $\nabla_t = \nabla G_t(f_t)$.*

When applying the lemma, we will often use the simple fact that

$$\| \nabla_t - M_t \|_* \| g_t - f_t \| = \inf_{\rho > 0} \left\{ \frac{\rho}{2} \| \nabla_t - M_t \|_*^2 + \frac{1}{2\rho} \| g_t - f_t \|^2 \right\} . \quad (3)$$

In particular, by setting $\rho = \eta$, we obtain the (unnormalized) regret bound of $\eta^{-1} R^2 + (\eta/2) \sum_{t=1}^{T} \| \nabla_t - M_t \|_*^2$, which is $R\sqrt{2 \sum_{t=1}^{T} \| \nabla_t - M_t \|_*^2}$ by choosing $\eta$ optimally. Since this choice is not known ahead of time, one may either employ the doubling trick, or choose the step size adaptively:

**Corollary 2.** *Consider step size $\eta_t = R_{\max} \min \left\{ \left( \sqrt{\sum_{i=1}^{t-1} \| \nabla_i - M_i \|_*^2} + \sqrt{\sum_{i=1}^{t-2} \| \nabla_i - M_i \|_*^2} \right)^{-1}, 1 \right\}$ with $R_{\max}^2 = \sup_{f,g \in \mathcal{F}} \mathcal{D}_{\mathcal{R}}(f, g)$. Then regret of the Optimistic Mirror Descent algorithm is upper bounded by $3.5 R_{\max} \left( \sqrt{\sum_{t=1}^{T} \| \nabla_t - M_t \|_*^2} + 1 \right) / T$.*

These results indicate that tighter regret bounds are possible if one can guess the next gradient $\nabla_t$ by computing $M_t$. One such case arises in *offline* optimization of a smooth function, whereby the previous gradient turns out to be a good proxy for the next one. More precisely, suppose we aim to optimize a function $G(f)$ whose gradients are Lipschitz continuous: $\| \nabla G(f) - \nabla G(g) \|_* \le H \| f - g \|$ for some $H > 0$. In this optimization setting, no guessing of $M_t$ is needed: we may simply query the oracle for the gradient and set $M_t = \nabla G(g_{t-1})$. The Optimistic Mirror Descent then becomes

$$f_t = \underset{f \in \mathcal{F}}{\arg\min} \ \eta_t \langle f, \nabla G(g_{t-1}) \rangle + \mathcal{D}_{\mathcal{R}}(f, g_{t-1}) \ , \ g_t = \underset{g \in \mathcal{F}}{\arg\min} \ \eta_t \langle g, \nabla G(f_t) \rangle + \mathcal{D}_{\mathcal{R}}(g, g_{t-1})$$

which can be recognized as the Mirror Prox method, due to Nemirovski [9]. By smoothness, $\|\nabla G(f_t) - M_t\|_* = \|\nabla G(f_t) - \nabla G(g_{t-1})\|_* \leq H\|f_t - g_{t-1}\|$. Lemma 1 with Eq. (3) and $\rho = \eta = 1/H$ immediately yields a bound

$$\sum_{t=1}^{T} G(f_t) - G(f^*) \leq HR^2,$$

which implies that the average $\bar{f}_T = \frac{1}{T} \sum_{t=1}^{T} f_t$ satisfies $G(\bar{f}_T) - G(f^*) \leq HR^2/T$, a known bound for Mirror Prox. We now extend this result to arbitrary $\alpha$-Hölder smooth functions, that is convex functions $G$ such that $\|\nabla G(f) - \nabla G(g)\|_* \leq H\|f - g\|^\alpha$ for all $f, g \in \mathcal{F}$.

**Lemma 3.** *Let $\mathcal{F}$ be a convex set in a Banach space $\mathcal{B}$ and let $\mathcal{R} : \mathcal{B} \to \mathbb{R}$ be a 1-strongly convex function on $\mathcal{F}$ with respect to some norm $\|\cdot\|$. Let $G$ be a convex $\alpha$-Hölder smooth function with constant $H > 0$ and $\alpha \in [0,1]$. Then the average $\bar{f}_T = \frac{1}{T} \sum_{t=1}^{T} f_t$ of the trajectory given by Optimistic Mirror Descent Algorithm enjoys*

$$G(\bar{f}_T) - \inf_{f \in \mathcal{F}} G(f) \leq \frac{8HR^{1+\alpha}}{T^{\frac{1+\alpha}{2}}}$$

*where $R \geq 0$ is such that $\sup_{f \in \mathcal{F}} \mathcal{D}_\mathcal{R}(f, g_0) \leq R$.*

This result provides a smooth interpolation between the $T^{-1/2}$ rate at $\alpha = 0$ (that is, no predictability of the gradient is possible) and the $T^{-1}$ rate when the smoothness structure allows for a dramatic speed up with a very simple modification of the original Mirror Descent.

## 3 Structured Optimization

In this section we consider the structured optimization problem

$$\underset{f \in \mathcal{F}}{\operatorname{argmin}} \, G(f)$$

where $G(f)$ is of the form $G(f) = \sup_{x \in \mathcal{X}} \phi(f, x)$ with $\phi(\cdot, x)$ convex for every $x \in \mathcal{X}$ and $\phi(f, \cdot)$ concave for every $f \in \mathcal{F}$. Both $\mathcal{F}$ and $\mathcal{X}$ are assumed to be convex sets. While $G$ itself need not be smooth, it has been recognized that the structure can be exploited to improve rates of optimization if the function $\phi$ is smooth [10]. From the point of view of online learning, we will see that the optimization problem of the saddle point type can be solved by playing two online convex optimization algorithms against each other (henceforth called Players I and II).

Specifically, assume that Player I produces a sequence $f_1, \ldots, f_T$ by using a regret-minimization algorithm, such that

$$\frac{1}{T} \sum_{t=1}^{T} \phi(f_t, x_t) - \inf_{f \in \mathcal{F}} \frac{1}{T} \sum_{t=1}^{T} \phi(f, x_t) \leq \operatorname{Rate}^1(x_1, \ldots, x_T) \tag{4}$$

and Player II produces $x_1, \ldots, x_T$ with

$$\frac{1}{T} \sum_{t=1}^{T} (-\phi(f_t, x_t)) - \inf_{x \in \mathcal{X}} \frac{1}{T} \sum_{t=1}^{T} (-\phi(f_t, x)) \leq \operatorname{Rate}^2(f_1, \ldots, f_T). \tag{5}$$

By a standard argument (see e.g. [7]),

$$\inf_f \frac{1}{T} \sum_{t=1}^{T} \phi(f, x_t) \leq \inf_f \phi(f, \bar{x}_T) \leq \sup_x \inf_f \phi(f, x)$$

$$\leq \inf_f \sup_x \phi(f, x) \leq \sup_x \phi(\bar{f}_T, x) \leq \sup_x \frac{1}{T} \sum_{t=1}^{T} \phi(f_t, x)$$

where $\bar{f}_T = \frac{1}{T} \sum_{t=1}^{T} f_t$ and $\bar{x}_T = \frac{1}{T} \sum_{t=1}^{T} x_t$. By adding (4) and (5), we have

$$\sup_{x \in \mathcal{X}} \frac{1}{T} \sum_{t=1}^{T} \phi(f_t, x) - \inf_{f \in \mathcal{F}} \frac{1}{T} \sum_{t=1}^{T} \phi(f, x_t) \leq \operatorname{Rate}^1(x_1, \ldots, x_T) + \operatorname{Rate}^2(f_1, \ldots, f_T) \tag{6}$$

which sandwiches the previous sequence of inequalities up to the sum of regret rates and implies near-optimality of $\bar{f}_T$ and $\bar{x}_T$.

**Lemma 4.** *Suppose both players employ the Optimistic Mirror Descent algorithm with, respectively, predictable sequences $M_t^1$ and $M_t^2$, 1-strongly convex functions $\mathcal{R}_1$ on $\mathcal{F}$ (w.r.t. $\|\cdot\|_{\mathcal{F}}$) and $\mathcal{R}_2$ on $\mathcal{X}$ (w.r.t. $\|\cdot\|_{\mathcal{X}}$), and fixed learning rates $\eta$ and $\eta'$. Let $\{f_t\}$ and $\{x_t\}$ denote the primary sequences of the players while let $\{g_t\}, \{y_t\}$ denote the secondary. Then for any $\alpha, \beta > 0$,*

$$\sup_{x \in \mathcal{X}} \phi\left(\bar{f}_T, x\right) - \inf_{f \in \mathcal{F}} \sup_{x \in \mathcal{X}} \phi(f, x) \tag{7}$$

$$\leq \frac{R_1^2}{\eta} + \frac{\alpha}{2} \sum_{t=1}^{T} \|\nabla_f \phi(f_t, x_t) - M_t^1\|_{\mathcal{F}^*}^2 + \frac{1}{2\alpha} \sum_{t=1}^{T} \|g_t - f_t\|_{\mathcal{F}}^2 - \frac{1}{2\eta} \sum_{t=1}^{T} \left( \|g_t - f_t\|_{\mathcal{F}}^2 + \|g_{t-1} - f_t\|_{\mathcal{F}}^2 \right)$$

$$+ \frac{R_2^2}{\eta'} + \frac{\beta}{2} \sum_{t=1}^{T} \|\nabla_x \phi(f_t, x_t) - M_t^2\|_{\mathcal{X}^*}^2 + \frac{1}{2\beta} \sum_{t=1}^{T} \|y_t - x_t\|_{\mathcal{X}}^2 - \frac{1}{2\eta'} \sum_{t=1}^{T} \left( \|y_t - x_t\|_{\mathcal{X}}^2 + \|y_{t-1} - x_t\|_{\mathcal{X}}^2 \right)$$

*where $R_1$ and $R_2$ are such that $\mathcal{D}_{\mathcal{R}_1}(f^*, g_0) \leq R_1^2$ and $\mathcal{D}_{\mathcal{R}_2}(x^*, y_0) \leq R_2^2$, and $\bar{f}_T = \frac{1}{T} \sum_{t=1}^{T} f_t$.*

The proof of Lemma 4 is immediate from Lemma 1. We obtain the following corollary:

**Corollary 5.** *Suppose $\phi : \mathcal{F} \times \mathcal{X} \mapsto \mathbb{R}$ is Hölder smooth in the following sense:*

$$\|\nabla_f \phi(f, x) - \nabla_f \phi(g, x)\|_{\mathcal{F}^*} \leq H_1 \|f - g\|_{\mathcal{F}}^{\alpha}, \quad \|\nabla_f \phi(f, x) - \nabla_f \phi(f, y)\|_{\mathcal{F}^*} \leq H_2 \|x - y\|_{\mathcal{X}}^{\alpha'}$$

$$and \quad \|\nabla_x \phi(f, x) - \nabla_x \phi(g, x)\|_{\mathcal{X}^*} \leq H_4 \|f - g\|_{\mathcal{F}}^{\beta}, \quad \|\nabla_x \phi(f, x) - \nabla_x \phi(f, y)\|_{\mathcal{X}^*} \leq H_3 \|x - y\|_{\mathcal{X}}^{\beta'}.$$

*Let $\gamma = \min\{\alpha, \alpha', \beta, \beta'\}$, $H = \max\{H_1, H_2, H_3, H_4\}$. Suppose both players employ Optimistic Mirror Descent with $M_t^1 = \nabla_f \phi(g_{t-1}, y_{t-1})$ and $M_t^2 = \nabla_x \phi(g_{t-1}, y_{t-1})$, where $\{g_t\}$ and $\{y_t\}$ are the secondary sequences updated by the two algorithms, and with step sizes $\eta = \eta' = (R_1^2 + R_2^2)^{\frac{1-\gamma}{2}} (2H)^{-1} \left(\frac{T}{2}\right)^{\frac{\gamma-1}{2}}$. Then*

$$\sup_{x \in \mathcal{X}} \phi\left(\bar{f}_T, x\right) - \inf_{f \in \mathcal{F}} \sup_{x \in \mathcal{X}} \phi(f, x) \leq \frac{4H(R_1^2 + R_2^2)^{\frac{1+\gamma}{2}}}{T^{\frac{1+\gamma}{2}}} \tag{8}$$

As revealed in the proof of this corollary, the negative terms in (7), that come from an upper bound on regret of Player I, in fact contribute to cancellations with positive terms in regret of Player II, and vice versa. Such a coupling of the upper bounds on regret of the two players can be seen as leading to faster rates under the appropriate assumptions, and this idea will be exploited to a great extent in the proofs of the next section.

## 4 Zero-sum Game and Uncoupled Dynamics

The notions of a zero-sum matrix game and a minimax equilibrium are arguably the most basic and important notions of game theory. The tight connection between linear programming and minimax equilibrium suggests that there might be simple dynamics that can lead the two players of the game to eventually converge to the equilibrium value. Existence of such simple or natural dynamics is of interest in behavioral economics, where one asks whether agents can discover static solution concepts of the game iteratively and without extensive communication.

More formally, let $A \in [-1, 1]^{n \times m}$ be a matrix with bounded entries. The two players aim to find a pair of near-optimal mixed strategies $(\bar{f}, \bar{x}) \in \Delta_n \times \Delta_m$ such that $\bar{f}^\top A \bar{x}$ is close to the minimax value $\min_{f \in \Delta_n} \max_{x \in \Delta_m} f^\top A x$, where $\Delta_n$ is the probability simplex over $n$ actions. Of course, this is a particular form of the saddle point problem considered in the previous section, with $\phi(f, x) = f^\top A x$. It is well-known (and follows immediately from (6)) that the players can compute near-optimal strategies by simply playing no-regret algorithms [7]. More precisely, on round $t$, the players I and II "predict" the mixed strategies $f_t$ and $x_t$ and observe $A x_t$ and $f_t^\top A$, respectively. While black-box regret minimization algorithms, such as Exponential Weights, immediately yield $\mathcal{O}(T^{-1/2})$ convergence rates, Daskalakis et al [6] asked whether faster methods exist. To make the problem well-posed, it is required that the two players are *strongly uncoupled*: neither $A$ nor the number of available actions of the opponent is known to either player, no "funny bit arithmetic" is allowed, and memory storage of each player allows only for constant number of payoff vectors. The authors of [6] exhibited a near-optimal algorithm that, if used by both players, yields a pair of

mixed strategies that constitutes an $\mathcal{O}\left(\frac{\log(m+n)(\log T+(\log(m+n))^{3/2})}{T}\right)$-approximate minimax equilibrium. Furthermore, the method has a regret bound of the same order as Exponential Weights when faced with an arbitrary sequence. The algorithm in [6] is an application of the excessive gap technique of Nesterov, and requires careful choreography and interleaving of rounds between the two non-communicating players. The authors, therefore, asked whether a simple algorithm (e.g. a modification of Exponential Weights) can in fact achieve the same result. We answer this in the affirmative. While a direct application of Mirror Prox does not yield the result (and also does not provide strong decoupling), below we show that a modification of Optimistic Mirror Descent achieves the goal. Furthermore, by choosing the step size adaptively, the same method guarantees the typical $\mathcal{O}(T^{-1/2})$ regret if not faced with a compliant player, thus ensuring robustness.

In Section 4.1, we analyze the "first-order information" version of the problem, as described above: upon playing the respective mixed strategies $f_t$ and $x_t$ on round $t$, Player I observes $Ax_t$ and Player II observes $f_t^\intercal A$. Then, in Section 4.2, we consider an interesting extension to partial information, whereby the players submit their moves $f_t, x_t$ but only observe the real value $f_t^\intercal A x_t$. Recall that in both cases the matrix $A$ is not known to the players.

## 4.1 First-Order Information

Consider the following simple algorithm. Initialize $f_0 = g_0' \in \Delta_n$ and $x_0 = y_0' \in \Delta_m$ to be uniform distributions, set $\beta = 1/T^2$ and proceed as follows:

```
On round t, Player I performs

      Play       f_t and observe Ax_t
      Update     g_t(i) ∝ g'_{t-1}(i) exp{−η_t[Ax_t]_i},    g'_t = (1 − β) g_t + (β/n) 1_n
                 f_{t+1}(i) ∝ g'_t(i) exp{−η_{t+1}[Ax_t]_i}

while simultaneously Player II performs

      Play       x_t and observe f_t^⊺A
      Update     y_t(i) ∝ y'_{t-1}(i) exp{−η'_t[f_t^⊺A]_i},    y'_t = (1 − β) y_t + (β/m) 1_m
                 x_{t+1}(i) ∝ y'_t(i) exp{−η'_{t+1}[f_t^⊺A]_i}
```

Here, $\mathbf{1}_n \in \mathbb{R}^n$ is a vector of all ones and both $[b]_i$ and $b(i)$ refer to the $i$-th coordinate of a vector $b$. Other than the "mixing in" of the uniform distribution, the algorithm for both players is simply the Optimistic Mirror Descent with the (negative) entropy function. In fact, the step of mixing in the uniform distribution is only needed when some coordinate of $g_t$ (resp., $y_t$) is smaller than $1/(nT^2)$. Furthermore, this step is also not needed if none of the players deviate from the prescribed method. In such a case, the resulting algorithm is simply the constant step-size Exponential Weights $f_t(i) \propto \exp\{-\eta \sum_{s=1}^{t-2}[Ax_{s-1}]_i + 2\eta[Ax_{t-1}]_i\}$, but with a factor 2 in front of the *latest* loss vector!

**Proposition 6.** *Let $A \in [-1,1]^{n\times m}$, $\mathcal{F} = \Delta_n$, $\mathcal{X} = \Delta_m$. If both players use above algorithm with, respectively, $M_t^1 = Ax_{t-1}$ and $M_t^2 = f_{t-1}^\intercal A$, and the adaptive step sizes*

$$\eta_t = \min\left\{\log(nT)\left(\sqrt{\sum_{i=1}^{t-1}\|Ax_i - Ax_{i-1}\|_*^2} + \sqrt{\sum_{i=1}^{t-2}\|Ax_i - Ax_{i-1}\|_*^2}\right)^{-1}, \frac{1}{11}\right\}$$

*and*

$$\eta_t' = \min\left\{\log(mT)\left(\sqrt{\sum_{i=1}^{t-1}\|f_i^\intercal A - f_{i-1}^\intercal A\|_*^2} + \sqrt{\sum_{i=1}^{t-2}\|f_i^\intercal A - f_{i-1}^\intercal A\|_*^2}\right)^{-1}, \frac{1}{11}\right\}$$

*respectively, then the pair $(\bar{f}_T, \bar{x}_T)$ is an $O\left(\frac{\log m + \log n + \log T}{T}\right)$-approximate minimax equilibrium. Furthermore, if only one player (say, Player I) follows the above algorithm, her regret against any sequence $x_1, \ldots, x_T$ of plays is*

$$\mathcal{O}\left(\frac{\log(nT)}{T}\left(\sqrt{\sum_{t=1}^{T}\|Ax_t - Ax_{t-1}\|_*^2} + 1\right)\right). \tag{9}$$

In particular, this implies the worst-case regret of $\mathcal{O}\left(\frac{\log(nT)}{\sqrt{T}}\right)$ in the general setting of online linear optimization.

We remark that (9) can give intermediate rates for regret in the case that the second player deviates from the prescribed strategy but produces "stable" moves. For instance, if the second player employs a mirror descent algorithm (or Follow the Regularized Leader / Exponential Weights method) with step size $\eta$, one can typically show stability $\|x_t - x_{t-1}\| = \mathcal{O}(\eta)$. In this case, (9) yields the rate $\mathcal{O}\left(\frac{\eta \log T}{\sqrt{T}}\right)$ for the first player. A typical setting of $\eta \propto T^{-1/2}$ for the second player still ensures the $\mathcal{O}(\log T/T)$ regret for the first player.

Let us finish with a technical remark. The reason for the extra step of "mixing in" the uniform distribution stems from the goal of having an adaptive and robust method that still attains $\mathcal{O}(T^{-1/2})$ regret if the other player deviates from using the algorithm. If one is only interested in the dynamics when both players cooperate, this step is not necessary, and in this case the extraneous $\log T$ factor disappears from the above bound, leading to the $O\left(\frac{\log n + \log m}{T}\right)$ convergence. On the technical side, the need for the extra step is the following. The adaptive step size result of Corollary 2 involves the term $R_{\max}^2 \geq \sup_g \mathcal{D}_{\mathcal{R}_1}(f^*, g)$ which is potentially infinite for the negative entropy function $\mathcal{R}_1$. It is possible that the doubling trick or the analysis of Auer et al [2] (who encountered the same problem for the Exponential Weights algorithm) can remove the extra $\log T$ factor while still preserving the regret minimization property. We also remark that $R_{\max}$ is small when $\mathcal{R}_1$ is instead the $p$-norm; hence, the use of this regularizer avoids the extraneous logarithmic in $T$ factor while still preserving the logarithmic dependence on $n$ and $m$. However, projection onto the simplex under the $p$-norm is not as elegant as the Exponential Weights update.

## 4.2 Partial Information

We now turn to the partial (or, zero-th order) information model. Recall that the matrix $A$ is not known to the players, yet we are interested in finding $\epsilon$-optimal minimax strategies. On each round, the two players choose mixed strategies $f_t \in \Delta_n$ and $x_t \in \Delta_m$, respectively, and observe $f_t^\top A x_t$. Now the question is, how many such observations do we need to get to an $\epsilon$-optimal minimax strategy? Can this be done while still ensuring the usual no-regret rate?

The specific setting we consider below requires that on each round $t$, the two players play four times, and that these four plays are $\delta$-close to each other (that is, $\|f_t^i - f_t^j\|_1 \leq \delta$ for $i, j \in \{1, \ldots, 4\}$). Interestingly, up to logarithmic factors, the fast rate of the previous section is possible even in this scenario, but we do require the knowledge of the number of actions of the opposing player (or, an upper bound on this number). We leave it as an open problem the question of whether one can attain the $1/T$-type rate with only one play per round.

| **Player I** | **Player II** |
|---|---|
| $u_1, \ldots, u_{n-1}$ : orthonormal basis of $\Delta_n$ | $v_1, \ldots, v_{m-1}$ : orthonormal basis of $\Delta_m$ |
| Initialize $g_1, f_1 = \frac{1}{n}\mathbf{1}_n$; Draw $i_0 \sim \text{Unif}([n-1])$ | Initialize $y_1, x_1 = \frac{1}{m}\mathbf{1}_m$; Draw $j_0 \sim \text{Unif}([m-1])$ |
| At time $t = 1$ to $T$ | At time $t = 1$ to $T$ |
|   **Play** $f_t$ |   **Play** $x_t$ |
|   Draw $i_t \sim \text{Unif}([n-1])$ |   Draw $j_t \sim \text{Unif}([m-1])$ |
|   **Observe :** |   **Observe :** |
|     $r_t^+ = (f_t + \delta u_{i_{t-1}})^\top A x_t$ |     $s_t^+ = -f_t^\top A(x_t + \delta v_{j_{t-1}})$ |
|     $r_t^- = (f_t - \delta u_{i_{t-1}})^\top A x_t$ |     $s_t^- = -f_t^\top A(x_t - \delta v_{j_{t-1}})$ |
|     $\bar{r}_t^+ = (f_t + \delta u_{i_t})^\top A x_t$ |     $\bar{s}_t^+ = -f_t^\top A(x_t + \delta v_{j_t})$ |
|     $\bar{r}_t^- = (f_t - \delta u_{i_t})^\top A x_t$ |     $\bar{s}_t^- = -f_t^\top A(x_t - \delta v_{j_t})$ |
|   **Build estimates :** |   **Build estimates :** |
|     $\hat{a}_t = \frac{n}{2\delta}\left(r_t^+ - r_t^-\right) u_{i_{t-1}}$ |     $\hat{b}_t = \frac{m}{2\delta}\left(s_t^+ - s_t^-\right) v_{j_{t-1}}$ |
|     $\bar{a}_t = \frac{n}{2\delta}\left(\bar{r}_t^+ - \bar{r}_t^-\right) u_{i_t}$ |     $\bar{b}_t = \frac{m}{2\delta}\left(\bar{s}_t^+ - \bar{s}_t^-\right) v_{j_t}$ |
|   **Update :** |   **Update :** |
|     $g_t(i) \propto g_{t-1}'(i) \exp\{-\eta_t \hat{a}_t(i)\}$ |     $y_t(i) \propto y_{t-1}'(i) \exp\{-\eta_t' \hat{b}_t(i)\}$ |
|     $g_t' = (1-\beta) g_t + (\beta/n)\mathbf{1}$ |     $y_t' = (1-\beta) y_t + (\beta/m)\mathbf{1}$ |
|     $f_{t+1}(i) \propto g_t'(i) \exp\{-\eta_{t+1} \bar{a}_t(i)\}$ |     $x_{t+1}(i) \propto y_t'(i) \exp\{-\eta_{t+1}' \bar{b}_t(i)\}$ |
| End | End |

**Lemma 7.** *Let $A \in [-1,1]^{n \times m}$, $\mathcal{F} = \Delta_n$, $\mathcal{X} = \Delta_m$, let $\delta$ be small enough (e.g. exponentially small in $m, n, T$), and let $\beta = 1/T^2$. If both players use above algorithms with the adaptive step sizes*

$$\eta_t = \min\left\{\sqrt{\log(nT)}\frac{\sqrt{\sum_{i=1}^{t-1}\|\hat{a}_i - \bar{a}_{i-1}\|_*^2} - \sqrt{\sum_{i=1}^{t-2}\|\hat{a}_i - \bar{a}_{i-1}\|_*^2}}{\|\hat{a}_{t-1} - \bar{a}_{t-2}\|_*^2}, \frac{1}{28m\sqrt{\log(mT)}}\right\}$$

*and*

$$\eta_t' = \min\left\{\sqrt{\log(mT)}\frac{\sqrt{\sum_{i=1}^{t-1}\|\hat{b}_i - \bar{b}_{i-1}\|_*^2} - \sqrt{\sum_{i=1}^{t-2}\|\hat{b}_i - \bar{b}_{i-1}\|_*^2}}{\|\hat{b}_{t-1} - \bar{b}_{t-2}\|_*^2}, \frac{1}{28n\sqrt{\log(nT)}}\right\}$$

*respectively, then the pair $(\bar{f}_T, \bar{x}_T)$ is an*

$$\mathcal{O}\left(\frac{\left(m\log(nT)\sqrt{\log(mT)} + n\log(mT)\sqrt{\log(nT)}\right)}{T}\right)$$

*-approximate minimax equilibrium. Furthermore, if only one player (say, Player I) follows the above algorithm, her regret against any sequence $x_1, \ldots, x_T$ of plays is bounded by*

$$\mathcal{O}\left(\frac{m\sqrt{\log(mT)}\log(nT) + n\sqrt{\log(nT)\sum_{t=1}^{T}\|x_t - x_{t-1}\|^2}}{T}\right)$$

We leave it as an open problem to find an algorithm that attains the $1/T$-type rate when both players only observe the value $e_i^\top A e_j = A_{i,j}$ upon drawing *pure* actions $i, j$ from their respective mixed strategies $f_t, x_t$. We hypothesize a rate better than $T^{-1/2}$ is not possible in this scenario.

## 5 Approximate Smooth Convex Programming

In this section we show how one can use the structured optimization results from Section 3 for approximately solving convex programming problems. Specifically consider the optimization problem

$$\underset{f \in \mathcal{G}}{\operatorname{argmax}} \quad c^\top f \tag{10}$$
$$\text{s.t.} \quad \forall i \in [d], \; G_i(f) \leq 1$$

where $\mathcal{G}$ is a convex set and each $G_i$ is an $H$-smooth convex function. Let the optimal value of the above optimization problem be given by $F^* > 0$, and without loss of generality assume $F^*$ is known (one typically performs binary search if it is not known). Define the sets $\mathcal{F} = \{f : f \in \mathcal{G}, c^\top f = F^*\}$ and $\mathcal{X} = \Delta_d$. The convex programming problem in (10) can now be reformulated as

$$\underset{f \in \mathcal{F}}{\operatorname{argmin}} \max_{i \in [d]} G_i(f) = \underset{f \in \mathcal{F}}{\operatorname{argmin}} \sup_{x \in \mathcal{X}} \sum_{i=1}^{d} x(i) G_i(f). \tag{11}$$

This problem is in the saddle-point form, as studied earlier in the paper. We may think of the first player as aiming to minimize the above expression over $\mathcal{F}$, while the second player maximizes over a mixture of constraints with the aim of violating at least one of them.

**Lemma 8.** *Fix $\gamma, \epsilon > 0$. Assume there exists $f_0 \in \mathcal{G}$ such that $c^\top f_0 \geq 0$ and for every $i \in [d]$, $G_i(f_0) \leq 1 - \gamma$. Suppose each $G_i$ is 1-Lipschitz over $\mathcal{F}$. Consider the solution*

$$\hat{f}_T = (1 - \alpha)\bar{f}_T + \alpha f_0$$

*where $\alpha = \frac{\epsilon}{\epsilon + \gamma}$ and $\bar{f}_T = \frac{1}{T}\sum_{t=1}^{T} f_t \in \mathcal{F}$ is the average of the trajectory of the procedure in Lemma 4 for the optimization problem (11). Let $\mathcal{R}_1(\cdot) = \frac{1}{2}\|\cdot\|_2^2$ and $\mathcal{R}_2$ be the entropy function. Further let $B$ be a known constant such that $B \geq \|f^* - g_0\|_2$ where $g_0 \in \mathcal{F}$ is some initialization and $f^* \in \mathcal{F}$ is the (unknown) solution to the optimization problem. Set $\eta = \underset{\eta \leq H^{-1}}{\operatorname{argmin}}\left\{\frac{B^2}{\eta} + \frac{\eta \log d}{1 - \eta H}\right\}$, $\eta' = \frac{1}{\eta} - H$, $M_t^1 = \sum_{i=1}^{d} y_{t-1}(i)\nabla G_i(g_{t-1})$ and $M_t^2 = (G_1(g_{t-1}), \ldots, G_d(g_{t-1}))$. Let number of iterations $T$ be such that*

$$T > \frac{1}{\epsilon} \inf_{\eta \leq H^{-1}}\left\{\frac{B^2}{\eta} + \frac{\eta \log d}{1 - \eta H}\right\}$$

We then have that $\hat{f}_T \in \mathcal{G}$ satisfies all $d$ constraints and is $\frac{\epsilon}{\gamma}$-approximate, that is

$$c^\top \hat{f}_T \geq \left(1 - \frac{\epsilon}{\gamma}\right) F^* \ .$$

Lemma 8 tells us that using the predictable sequences approach for the two players, one can obtain an $\frac{\epsilon}{\gamma}$-approximate solution to the smooth convex programming problem in number of iterations at most order $1/\epsilon$. If $T_1$ (reps. $T_2$) is the time complexity for single update of the predictable sequence algorithm of Player I (resp. Player 2), then time complexity of the overall procedure is $\mathcal{O}\left(\frac{T_1 + T_2}{\epsilon}\right)$

## 5.1 Application to Max-Flow

We now apply the above result to the problem of finding Max Flow between a source and a sink in a network, such that the capacity constraint on each edge is satisfied. For simplicity, consider a network where each edge has capacity $1$ (the method can be easily extended to the case of varying capacity). Suppose the number of edges $d$ in the network is the same order as number of vertices in the network. The Max Flow problem can be seen as an instance of a convex (linear) programming problem, and we apply the proposed algorithm for structured optimization to obtain an approximate solution.

For the Max Flow problem, the sets $\mathcal{G}$ and $\mathcal{F}$ are given by sets of linear equalities. Further, if we use Euclidean norm squared as regularizer for the flow player, then projection step can be performed in $\mathcal{O}(d)$ time using conjugate gradient method. This is because we are simply minimizing Euclidean norm squared subject to equality constraints which is well conditioned. Hence $T_1 = \mathcal{O}(d)$. Similarly, the Exponential Weights update has time complexity $\mathcal{O}(d)$ as there are order $d$ constraints, and so overall time complexity to produce $\epsilon$ approximate solution is given by $\mathcal{O}(nd)$, where $n$ is the number of iterations of the proposed procedure.

Once again, we shall assume that we know the value of the maximum flow $F^*$ (for, otherwise, we can use binary search to obtain it).

**Corollary 9.** *Applying the procedure for smooth convex programming from Lemma 8 to the Max Flow problem with $f_0 = \mathbf{0} \in \mathcal{G}$ the $0$ flow, the time complexity to compute an $\epsilon$-approximate Max Flow is bounded by*

$$\mathcal{O}\left(\frac{d^{3/2}\sqrt{\log d}}{\epsilon}\right) \ .$$

This time complexity matches the known result from [8], but with a much simpler procedure (gradient descent for the flow player and Exponential Weights for the constraints). It would be interesting to see whether the techniques presented here can be used to improve the dependence on $d$ to $d^{4/3}$ or better while maintaining the $1/\epsilon$ dependence. While the result of [5] has the improved $d^{4/3}$ dependence, the complexity in terms of $\epsilon$ is much worse.

## 6 Discussion

We close this paper with a discussion. As we showed, the notion of using extra information about the sequence is a powerful tool with applications in optimization, convex programming, game theory, to name a few. All the applications considered in this paper, however, used some notion of smoothness for constructing the predictable process $M_t$. An interesting direction of further research is to isolate more general conditions under which the next gradient is predictable, perhaps even when the functions are not smooth in any sense. For instance one could use techniques from bundle methods to further restrict the set of possible gradients the function being optimized can have at various points in the feasible set. This could then be used to solve for the right predictable sequence to use so as to optimize the bounds. Using this notion of selecting predictable sequences one can hope to derive adaptive optimization procedures that in practice can provide rapid convergence.

**Acknowledgements:** We thank Vianney Perchet for insightful discussions. We gratefully acknowledge the support of NSF under grants CAREER DMS-0954737 and CCF-1116928, as well as Dean's Research Fund.

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
