[Reviews · NeurIPS 2013]

Submitted by Assigned_Reviewer_2

The authors apply the Optimistic Mirror Descent algorithm to a variety of problems where smoothness induces a certain amount of predictability in the resulting loss function sequences. In particular, better rates are possible when either: 1) the objective G is fixed for all t, and smooth, or 2) the problem is a saddle point problem on a function phi(f, x) which is appropriately smooth for each of its arguments individually, and the optimistic mirror descent algorithm is played against itself. These are rather specialized notions of predictability, but they lead to some interesting results. Section 4 extends this to zero-sum games, and Section 5 considers approximate smooth convex programming.

Unlike many applications of online algorithms, the goal in most cases in this paper is to solve a batch optimization problem. This should probably be made more explicit in the introduction.

Section 3: The author’s need to be more clear about why this section improves on the result from Lemma 3. The key point (I think) is that the smoothness assumptions of Cor 5 may be satisfied even if G(f) = inf_x phi(f, x) is not smooth. Note that an alternative approach to solving the problem of Sec 3 is to run an online algorithm against a best-response adversary. It is worth asking whether or not any online algorithm paired with a best-response oracle can lead to the convergence rates like 1/T produced by the procedure introduced here. (Perhaps not, since this would appear to destroy predictability).

Section 4: Be more clear on how this section improves on Section 3. The key seems to be adding robustness to an arbitrary adversary in addition to the good bounds for the predictable one. The results of Sec 4.2 are rather weaker than the section title suggests. I would suggest describing this setting as multi-point bandit feedback, and citing “Optimal Algorithms for Online Convex Optimization with Multi-Point Bandit Feedback”. Based on the notational similarities, it seems quite likely the authors are aware of this work. It really should have been cited in the submission. It also seems somewhat surprising that 4 points are needed, rather than just 2 --- this should be discussed.

Section 5: It’s not clear to me that one can assume F^* is known WLOG. At the very least, the binary search procedure needs to be made precise: Are we running the algorithm of section 5 at each step of the binary search? If so, how do you check whether the guess of F^* is too high or too low? How does using a guess of F^* that is off by some amount delta impact the final bound of Lemma 8? This carries over to Sec 5.1 as well of course.

Also, Eq (11) appears to fit into the setup of Sec 3 (Cor 5), why do we need the machinery of Sec 4.2 here (Lemma 7)? Is Line 368 a typo?

In general, the paper suffers from some typos that made it more difficult to review, and could also probably use a bit more discussion of related work (several particular citations are recommended below).

Local points:

- Line 112: This simple intuition and analysis for Mirror Prox is very nice.

- Line 93: This bound can be seen as a direct generalization of the result that the “Be The Leader” (BTL) algorithm suffers no regret, since you arrive at a BTL algorithm by taking Delta_t = M_t. This is worth mentioning, as well as a citation to “Efficient algorithms for online decision problems” (Adam Kalai and Santosh Vempala).

- Line 131: I would find the section title “Saddle-Point Optimization” more clear.

- Line 154-155: citing “Adaptive game playing using multiplicative weights” (Yoav Freund and Robert E. Schapire) is probably appropriate. This citation is also appropriate at lines 208-209.

- Line 223: More detail needed, explain exactly why Sec 3 doesn’t lead to this result (because it doesn’t handle an arbitrary adversary as well?), as well as why a direct application of mirror-prox does not provide strong decoupling.

- Line 225, also lines 265 and 268: It’s imprecise to refer to regret bounds like 1/T or 1/Sqrt(T), you need to multiply through by T to get regret as defined on line 65. Alternatively, refer to these as convergence rates.

- Line 312-314: Typos in both “Build estimates” sections of the pseudo-code, you want r^+ - r^-, as written all the a’s and b’s are zero.

- Line 366: Epsilon hasn’t be defined yet, introduce as an arbitrary constant eps > 0 earlier in the statement.

- Line 368: Do you mean Corollary 5, not Lemma 7? If Lemma 7 is really intended, then the procedure needs to be made more clear, e.g., what is the matrix A here?

- Appendix: The proofs make extensive use of copy-paste, which makes them much harder to read because there is so much repetition. Please re-structure the proofs so they are less verbose and the main arguments are more clear.
Summary: The paper analyzes some new applications of Optimistic Mirror Descent, in particular, a simple analysis and intuition for fixed smooth functions, and an analysis applicable to self-play of the algorithm. The results seem novel and interesting.

Submitted by Assigned_Reviewer_5

Optimization, Learning, and Games with Predictable Sequences

Summary:
The paper discusses Optimistic Mirror Descent (OMD). This is a variant of the classical algorithm for online convex optimisation that is especially good when the sequence of loss function gradients exhibits some predictable regularity that can be exploited by the learner.

The authors first give the specific substitution for which OMD reduces to Mirror Prox, and generalise the known bounds to Hoelder smooth functions. The paper then proceeds with the analysis of objectives with saddle-point structure. It first discusses plain min-max optimisation by a single algorithm. It then turns to a game-theoretic viewpoint, and discusses how two opponents can arrive at a saddle point faster when both players use the proposed algorithm, while maintaining the worst-case sqrt(T) regret without cooperation. The paper concludes by considering approximate convex optimisation with an application to approximate max flow.

Discussion:

This paper on prediction of predictable sequences is well executed and rich. The exposition is very clear, and the results are strong. I very much like the application to saddle-point finding, where the players cannot assume cooperation but there is a speedup if it happens.



Small things:
Page 5, proposition 6. In your notation you access x_0 and f_0, but these are undefined.
page 7, (11). Here x[i] should be x_i.
Summary: This beautiful paper investigates prediction with predictable sequences (of gradients) and obtains strong results with interesting technical applications.

Submitted by Assigned_Reviewer_6

The paper provides applications of online learning with predictable loss sequences. First earlier results are generalized to Holder-smooth loss functions. The framework is then applied to compute solutions to saddle-point type problems, as well as to efficient computation of equilibria in two-player zero-sum games (for cooperative players, with guarantees if one of the players does not cooperate). Finally, convex programming problems are considered, but here I find the underlying assumptions strange.

I think the problems considered are of natural interest, and the approach is novel. It is somewhat misleading though, in some sense, that the "predictable sequence framework" uses, as prediction, past observations, and smoothness to quantify how good the predictions are.

The paper is generally well-written and clear. However, I have two major criticisms which need to be addressed in the authors' response:

1. All learning rates in the paper are set with peeking into the future: \eta_t is a function of \Nabla_t, which only becomes available after an update with \eta_t is performed, and the next gradient (loss) \Nabla_t is revealed.

2. Section 5: I do not understand the assumption that F^* can be known in advance, or can be found using binary search. Why would verifying the existence of a solution with a given value be any easier than actually finding such a solution? So why is it a valid assumption to know the exact (not approximate) value of the maximum without actually knowing an optimal solution? (In which case the whole following optimization process is pointless.)


Minor comments:
- 1. It does not seem to be necessary to give the proof of Lemma 1, copied verbatim from [9], except for the unproven first line in the statement. Instead, Lemma 3 of [9] should be referred to.

- 2. Proof of Lemma 2: line571: it seems that 1/eta_1 \le 1/R_max is implicitly assumed, which is the same as assuming the norm of the gradient is bounded by 1. Please make this explicit (or avoid this bound).

- 3. line 11: It seems that \rho should be 1/\sqrt{H} to cancel the first term in (3), deteriorating the bound in line 113 worse.

- 4. Explain the inequalities in the derivations, such as lines 587 and 590, 637, 645, 655, 659, etc. These are not necessarily hard, but sometimes simply the inequalities are so long that it is hard to find the differences...

- 5. Corollary 5: define the norms _F and _X

- Proposition 6: equation (9) bounds the average (or normalized) regret. The same for lemma 7.

The authors might also be interested in the recent ICML paper Dynamical Models and tracking regret in online convex programming by Eric Hall and Rebecca Willett (JMLR W&CP 28(1):579-587, 2013), which considers predictability in a related, albeit somewhat different setting.

ADDED AFTER THE REBUTTAL PERIOD:
- learning rates: the corrections seems ok
- Section 5: knowing F^* in advance: as I understand the response, F^* can only be known approximately, and this approximation has some price in the bounds. So I would not say knowing F^* can be assumed without loss of generality. However, the explanation given would be sufficient.
- bound in line 113: you are right, it was my mistake.
Summary: I think this is a solid paper with several interesting results.
Author Feedback

Author rebuttal: We thank the reviewers for the detailed comments. Let us address the particular points raised in the reviews:

Reviewer 1:

** Section 3: The author’s need to be more clear about why this section improves on the result from Lemma 3.

As the reviewer correctly points out, the key observation in Section 3 is that faster rates may be obtained even for non-smooth functions G that can be represented in the form G = inf_x phi(f,x) with smooth \phi. A standard example is the l1-norm function. We will make this point more clear.

** Section 4: Be more clear on how this section improves on Section 3.

Indeed, the key to Section 4 is ``robustness'' with respect to adversarial sequences vs benign ones. This robustness is achieved through adaptive step size. The reason we need 4 queries is that 2 are used for the predictable sequence and 2 for the current loss. We do not know whether this is necessary.

** Section 5: It’s not clear to me that one can assume F^* is known WLOG.

Two reviewers asked about this, so let us clarify. Irrespective of what value of F* we choose, the solution f^_T returned by the algorithm will have the objective value of (1- epsilon /gamma)F*. However, if the problem with (1- epsilon /gamma)F* is infeasible (that is, this value is larger than OPT), the solution will fail at least one of the constraints. Thus, to check whether an objective value is larger than OPT or not, we simply run the proposed algorithm for the prescribed number of rounds T (which only depends on epsilon and not F*) and at the end of the T rounds check the constraints. This way we may perform binary search. Finally one obtains an approximate solution by paying only an extra log(1/epsilon) factor. This is a standard step (see e.g. [5]) and we will make this explanation more explicit in the text. Further, and importantly, making gradient steps and projection on F is cheaper than solving the original problem.

** Eq (11) appears to fit into the setup of Sec 3 (Cor 5), why do we need the machinery of Sec 4.2 here (Lemma 7)? Is Line 368 a typo?

Yes, we thank the reviewers for pointing this out, this is a typo. We meant to refer to Lemma 4.

** Line 223: More detail needed, explain exactly why Sec 3 doesn’t lead to this result (because it doesn’t handle an arbitrary adversary as well?), as well as why a direct application of mirror-prox does not provide strong decoupling.

Note that the predictable sequence in mirror prox is not something that player 1 observes: it is some internal knowledge known only to player 2. On the other hand the strategy we propose only uses the observed payoff vector from round t-1 as the predictable sequence. Second justification is that variable step-size is needed for adaptivity to any adversary while retaining fast rate versus honest protocol.




Reviewer 3:

** All learning rates in the paper are set with peeking into the future: \eta_t is a function of \Nabla_t, which only becomes available after an update with \eta_t is performed, and the next gradient (loss) \Nabla_t is revealed.

We thank the reviewer for noticing this! The result is still correct, but eta_t should be replaced by eta_{t-1}, as is currently written in the bandit version. The proof only needs the following minor edits :

- When both players are honest, exactly the same proof works since the bound for eta_t^-1 used there holds for eta_{t-1}^-1 as well
- When opponent is adversarial, we get an extra telescoping sum after applying Holder with eta_t. That is, only the last 2 lines of proof change and we get instead :

sum_t (eta_t^{-1} -eta_{t-1}^{-1}) ||f_t -g'_t||^2 \leq R^2 sum_t (eta_t^{-1} -eta_{t-1}^{-1})
\leq R^2 (eta_T^{-1} -eta_1^{-1})

Hence, overall we get an extra factor of 2 in the bound. Similar rectification needs to be made for corollary 2 as well. These changes will be reflected in the final version.

** Section 5: I do not understand the assumption that F^* can be known in advance

Please see the response to reviewer 1.

** line 11: It seems that \rho should be 1/\sqrt{H} to cancel the first term in (3), deteriorating the bound in line 113 worse.

We believe the statement is correct as written. Notice that H becomes squared, which was perhaps missed by the reviewer.